# Combination of Lanosterol and Nilvadipine Nanosuspensions Rescues Lens Opacification in Selenite-Induced Cataractic Rats

**DOI:** 10.3390/pharmaceutics14071520

**Published:** 2022-07-21

**Authors:** Saori Deguchi, Reita Kadowaki, Hiroko Otake, Atsushi Taga, Yosuke Nakazawa, Manju Misra, Naoki Yamamoto, Hiroshi Sasaki, Noriaki Nagai

**Affiliations:** 1Faculty of Pharmacy, Kindai University, 3-4-1 Kowakae, Higashi-Osaka 577-8502, Osaka, Japan; 2045110002h@kindai.ac.jp (S.D.); kadowaki.reita@kindai.ac.jp (R.K.); hotake@phar.kindai.ac.jp (H.O.); punk@phar.kindai.ac.jp (A.T.); 2Faculty of Pharmacy, Keio University, 1-5-30 Shibakoen, Minato-ku 105-8512, Tokyo, Japan; nakazawa-ys@pha.keio.ac.jp; 3Graduate School of Pharmacy, Gujarat Technological University Gandhinagar Campus Nr. Government Polytechnic K-6 Circle, E-4 Electronic Estate G.I.D.C, Sector-26, Gandhinaga 382028, Gujarat, India; asso_manju_misra@gtu.edu.in; 4Laboratory of Molecularbiology and Histochemistry, Fujita Health University Institute of Joint Research, 1-98 Dengakugakubo, Kutsukake, Toyoake 470-1192, Aichi, Japan; naokiy@fujita-hu.ac.jp; 5Department of Ophthalmology, Kanazawa Medical University, 1-1 Daigaku, Uchinada, Kahoku 920-0293, Ishikawa, Japan; mogu@kanazawa-med.ac.jp

**Keywords:** cataract, lanosterol, nilvadipine, nanosuspensions, selenite-induced cataractic rat

## Abstract

It has recently been reported that lanosterol (LAN) plays a preventive role against lens opacification through the reversal of crystalline aggregation. However, the effect of LAN is not sufficient to restore lens transparency. In this study, we designed ophthalmic nanosuspensions (LAN-ONSs and NIL-ONSs) based on LAN and nilvadipine (NIL), which can counteract cataract-related factors (e.g., enhanced Ca^2+^ and calpain levels), and investigated whether the combination of LAN-ONSs and NIL-ONSs can restore the nuclear lens opacity in sodium-selenite-induced cataractic rats (cataractic rats). The mean particle sizes of the LAN-ONSs and NIL-ONSs were 108.8 nm and 89.0 nm, respectively. The instillation of the LAN-ONSs or NIL-ONSs successfully delivered the drugs (LAN or NIL) into the lenses of the rats, although the instillation of LAN-ONSs or NIL-ONSs alone did not increase lens transparency in the cataractic rats. On the other hand, the cataract-related factors (enhanced Ca^2+^ and calpain levels) were significantly alleviated by the combination of LAN-ONSs and NIL-ONSs; furthermore, the perinuclear refractile ring in the lens nucleus and enhanced number of swollen fibers were attenuated by the LAN-ONS and NIL-ONS combination. Moreover, the opacity levels in the cataractic rats were reduced after treatment with the combination of LAN-ONSs and NIL-ONSs. It is possible that the combination of LAN and NIL will be useful for the treatment of lens opacification in the future.

## 1. Introduction

Cataracts are one of the most common causes of blindness worldwide if left untreated. Moreover, it is anticipated that the incidence of cataracts will continue to increase with increasing lifespan [1,2]. A cataract can be classified as a cortical cataract, nuclear cataract, or posterior sub-capsular cataract, according to the region of opacity [3]. The common feature of cataractogenesis is a multi-factorial process involving the aggregation of misfolded crystalline proteins [4]. Cataracts are characterized by the activation of m-calpain, apoptosis of lens epithelial cells, lipid peroxidation, reduction of endogenous antioxidants, and reactive oxygen species production [5,6]. At present, surgery is the only established treatment, involving surgical removal of the opacified lens. However, despite progress in surgical techniques, this is still sub-optimal in developing countries, and cataracts remain the leading cause of visual impairment in developing countries, due to the lack of access to eye care in the developing world [7,8]. In addition, double vision, cystoid macular edema, detached retina, and posterior capsule opacification may also be observed as complications in patients treated using surgical techniques [7]. Therefore, strategies to prevent the development of opacification and complications associated with surgery, as well as to maintain lens transparency, call for alternative cataract treatments instead of surgical techniques.

The previous studies of Zhao et al. [9] and Makley et al. [10] have reported that lanosterol (LAN) and 25-hydroxycholesterol can redissolve crystallin aggregates in animal lenses, partially restoring lens transparency. Thus, the paradigm that vision can only be restored through cataract surgery has recently been challenged. We previously designed ophthalmic nanosuspensions of LAN (LAN-ONSs) and reported that repeated instillation of LAN-ONSs attenuated the slight collapse of the lens structure in Shumiya cataract rats (SCR), although the serious structural collapses were not reversed with the instillation of LAN-ONSs, and repeated instillation could not reverse or stop lens opacification [11]. These results suggest that this discrepancy may be caused by the difference in the period and dose of LAN, or by differences in the models used. In addition, in order to restore opaque lenses to transparency using LAN-ONSs, it is important not only to cure the collapse of the tissue structure, but also to suppress other cataract-related factors.

It is well-known that most cataracts present with an increase in calcium ion (Ca^2+^) concentration in the lens [12], where the increased Ca^2+^ contents and the subsequent stimulation of calpain activity cause the onset of cataract formation [13,14,15,16]. The regulation of intracellular Ca^2+^ at low levels serves to maintain the stability of lens proteins. Our previous reports have also shown that the instillation of ophthalmic nanosuspensions of nilvadipine (NIL-ONSs), a dihydropyridine L-type voltage-dependent Ca^2+^ channel (VDCC) blocker, could delay the onset of cataract development in SCR [17]. Therefore, repairing the lens tissue structure with LAN-ONSs while controlling the Ca^2+^ concentration in the lens with NIL-ONSs may be useful for the maintenance or restoration of lens transparency.

The selenite cataract is an animal model of cataracts produced in suckling rat pups [18,19,20,21,22,23] that presents with calpain-induced proteolysis, suppression of mitosis, decreased rate of epithelial cell differentiation, and cytoskeletal loss [14,24]. These characteristics of the model animals are similar to those seen clinically in humans, and this model has widely been used in many studies as a useful in vivo model for the initial screening of potential anti-cataract agents. In this study, we investigate whether the combination of LAN-ONSs and NIL-ONSs can restore the lens opacification in selenite-induced cataractic rats.

## 2. Materials and Methods

### 2.1. Animals

Male Wistar rat pups aged 11 days old were used and were housed with a controlled lighting schedule (7:00–19:00 light; 19:00–7:00 dark) at 25 °C. They were provided unlimited access to drinking water and a CE-2 formulation diet (Clea Japan Inc., Tokyo, Japan). The instillation was started in 18-day-old rats with or without mature nuclear cataracts, and 5 µL of ophthalmic formulation was repeatedly instilled into the right eye for 28 days (35 days after the injection of sodium selenite). All experiments were performed in accordance with the guidelines for the Association for Research in Vision and Ophthalmology (ARVO) and the Pharmacy Committee Guidelines for the Care and Use of Laboratory Animals. Moreover, the experiments using animals were approved on 1 April 2021 (project identification code, KAPS-2021-004), by Kindai University.

### 2.2. Chemicals

NIL powder, propyl p-hydroxybenzoate, Ca Test Kits, and mannitol were purchased from Wako Pure Chemical Industries, Ltd. (Osaka, Japan). LAN powder was obtained from Nacalai Tesque Inc. (Kyoto, Japan). Benzalkonium chloride (BAC) was provided by Kanto Chemical Co., Inc. (Tokyo, Japan). Pivalephrine (0.1%) was purchased from Santen Pharmaceutical Co., Ltd. (Osaka, Japan). Bio-Rad Protein Assay Kit was provided by Bio-Rad Laboratories (Hercules, CA, USA), and 2-hydroxypropyl-β-cyclodextrin (HPβCD) was obtained from Nihon Shokuhin Kako Co., Ltd. (Tokyo, Japan). Type SM-4 of methylcellulose (MC) and hydroxypropyl-methylcellulose (HPMC) were supplied by Shin-Etsu Chemical Co., Ltd. (Tokyo, Japan). All other chemicals were of the highest purity commercially available.

### 2.3. Preparation of LAN-ONSs and NIL-ONSs

The LAN-ONSs and NIL-ONSs were prepared following our previous reports [11,17], and their compositions are detailed in Table 1. Briefly, the LAN-ONSs and NIL-ONSs were prepared by using a bead mill method. For the preparation of LAN-ONSs, the LAN powder was dispersed in purified water containing HPβCD, MC, BAC, and mannitol, and milled with 0.1 mm zirconia beads using a Bead Smash 12 (Wakenyaku Co. Ltd., Kyoto, Japan) at 5500 rpm for 30 s at 4 °C. The milling treatment was repeated 20 times. Then, purified water containing HPβCD, MC, BAC, mannitol, and the dispersions was gently stirred to eliminate the air bubbles generated by the bead mill. After the disappearance of air bubbles, the mill process using the Bead Smash 12 was carried out as before (5500 rpm, 30 s × 20 times, 4 °C), and the milled dispersions were used as LAN-ONSs. The NIL-ONSs were prepared as follows: a mixture of NIL powder, MC, and mannitol was milled in an agate mortar for 60 min, then crushed with 1 mm zirconia beads using a Bead Smash 12 at 3000 rpm for 30 s at 4 °C. Next, the mixture was transferred into a tube containing purified water with HPβCD and milled with 0.1 mm zirconia beads using the Bead Smash 12 (5500 rpm, 30 s × 30 times, 4 °C). Subsequently, the mixture was treated in a Shake Master NEO (1500 rpm, 1 h × 3 times, 4 °C), then used as NIL-ONSs. These ophthalmic formulations were sterilized using 220 nm filters.

### 2.4. Measurement of LAN and NIL

The LAN concentration was measured, according to a previous study, by using a simple liquid chromatography (LA)-charged aerosol detector (CAD) method [11,25]. Twenty microliters of samples in methanol were injected, and the LAN in samples was separated by an LC-10AD pump (Shimadzu, Kyoto, Japan) with TSK gel ODS-100S (Tosoh Co., Tokyo, Japan). A step gradient experiment was performed using 5% and 100% methanol over 19 min (flow rate, 0.4 mL/min), and the LAN was detected using a Corona Veo detector (Thermo Fisher Scientific, Inc., Waltham, MA, USA). The charger voltage, charger current, and inlet pressure (nitrogen) were 2.66 kV, 0.99 µA, and 61.9 psi, respectively.

The NIL concentration was measured with an HPLC method consisting of a Shimadzu model LC-20AT pump, a Shimadzu degasser model DGU-20A, an auto-sampler SIL-10AF, a column oven CTO-20A, and a UV detector SPD-20A (HPLC, Shimadzu Corp., Kyoto, Japan). The samples (10 μL) were injected by the SIL-10AF, and the columns used were Inertsil^®^ ODS-3 columns (GL Science Co., Inc., Tokyo, Japan). The mobile phase consisted of 50 mM phosphate buffer/acetonitrile/methanol (50/25/25, *v*/*v*%) at a flow rate of 0.25 mL/min, the column temperature was 35 °C, and the wavelength for detection were selected as 242 nm. One microgram per milliliter propyl p-hydroxybenzoate was used as an internal standard for NIL measurement [17].

### 2.5. Evaluation of Drug Particles in LAN-ONSs and NIL-ONSs

The characteristics of the ophthalmic formulations were measured following our previous reports [11,17]. Dynamic light scattering with NANOSIGHT LM10 (QuantumDesign Japan, Tokyo, Japan) was used to measure the particle size distribution, and the atomic force microscope (AFM) images of LAN-ONSs and NIL-ONSs were obtained with a scanning probe microscope SPM-9700 (Shimadzu Corp., Kyoto, Japan).

### 2.6. Measurement of Drug Content in the Lenses

Twenty microliters of ophthalmic formulations (0.5% LAN or 0.6% NIL), shown in Table 1, were single-instilled into the right eye of 18-day-old rats (*n* = 8), and the eyes were kept open for about 1 min to prevent the ophthalmic formulations from being washed out. After 3, 6, and 24 h, the lenses of the rats were removed and homogenized in 200 µL methanol. The homogenates were centrifuged at 20,400× *g* for 15 min at 4 °C, and the concentrations of LAN and NIL in the supernatants were measured by the LA-CAD and HPLC methods described above. In this study, two formulations (0.5% LAN and 0.6% NIL) were instilled to same rats, and the drug content in the lenses was measured. Moreover, the LAN contents in the lenses of rats instilled with LAN were calculated as the difference from that in the lenses of non-instilled rats (1.47 ± 0.11 nmol/lens; *n* = 5), since the LAN is also contained in the normal lens.

### 2.7. Assessment of Lens Opacification via Imaging

At the onset of cataract formation, the 11-day-old rats were given a single subcutaneous injection of sodium selenite dissolved in saline (19 µmol/kg). Twenty microliters of ophthalmic formulations (0.5% LAN or 0.6% NIL), shown in Table 1, were instilled one time/day for 28 days. The start day and the end day of dosing were 7 and 35 days after injection of sodium selenite, respectively. Digital camera images were captured with a cover glass placed over the rat’s eyes under surface anesthesia (0.4% Benoxil). On the other hand, for the measurement of Scheimpflug slit images, the rats without anesthesia were dilated through the instillation of 0.1% pivalephrine and monitored by digital camera and an EAS-1000 equipped with a CCD camera (Nidek, Gamagori, Japan) [11,17]. Transparency of the lens was calculated as follows: the outline of the lens image was determined by selecting four points on the image, and then the transparent area within the outline and thread level were set automatically by the software. The total area of opacity, in pixels, was analyzed by a computer using image analysis software connected to the EAS-1000 system. The parameters of the EAS-1000 were as follows: thread level, 100; flash level, 100; and slit length, 4.2 mm.

### 2.8. Evaluation of Cataract-Related Factors

At the onset of cataract formation, the 11-day-old rats were given a single subcutaneous injection of sodium selenite dissolved in saline (19 µmol/kg). Five microliters of ophthalmic formulation were instilled into the right eye of 18-day-old rats (normal or sodium-selenite-induced cataractic rats), which were euthanized by injection of a lethal dose of sodium pentobarbital 3 h after instillation. After that, the lenses of the rats were removed and homogenized in phosphate-buffered saline (pH 7.4) on ice. The lens homogenates were centrifuged at 20,400× *g* for 30 min at 4 °C, and the supernatants were used for the measurement of cataract-related factors, such as Ca^2+^–ATPase activity, Ca^2+^ content, and calpain activity. The Ca^2+^–ATPase activity was calculated following our previous report [26]. Briefly, the samples were incubated in solution A (pH 7.4; water, 100 mM HEPES, 200 mM KCl, 2 mM EGTA, 10 mM MgCl_2_, and 2 mM ATP) with or without 2.2 mM CaCl_2_ for 1 h at 37 °C. Then, the reaction was stopped using trichloroacetic acid and calculated as the difference in the Pi liberated from ATP measured in the presence and absence of Ca^2+^ to determine the absorbance of supernatants, which was measured at 660 nm. In addition, a Ca Test Kit, according to methyl xylenol blue colorimetric method, was used to measure the Ca^2+^ content [27], which is expressed as µmol/mg protein. The calpain activity was analyzed by using a Calpain Activity Fluorometric Assay Kit, and Abs (505 nm) was measured, according to the manufacturer’s instructions [11]. The protein levels in samples used to determine Ca^2+^–ATPase activity and Ca^2+^ content were determined according to the Bradford method using a Bio-Rad Protein Assay Kit.

### 2.9. Evaluation of Lens Structure in Selenite-Induced Cataractic Rats Using Hematoxylin and Eosin (H.E.) Staining

At the onset of cataract formation, the 11-day-old rats were given a single subcutaneous injection of sodium selenite dissolved in saline (19 µmol/kg). Twenty microliters of ophthalmic formulations (0.5% LAN or 0.6% NIL), shown in Table 1, were instilled one time/day for 28 days. The start day and the end day of dosing were 7 and 35 days after injection of sodium selenite, respectively. H.E. staining was performed following our previous reports [11,25]. Normal and cataractic rats (35 days after injection of sodium selenite) were euthanized by injecting a lethal dose of pentobarbital, and the eyes were removed and fixed at room temperature for 2 days using SUPER FIX. Three micrometer paraffin serial sections were prepared by microtome, the fixed lenses were prepared in paraffin blocks, and H.E. staining was performed for morphological observation. A biological upright microscope (Power BX-51, Olympus, Tokyo, Japan) was used to observe the specimens.

### 2.10. Statistical Analysis

The statistical analysis was performed by using Student’s *t*-test and ANOVA, followed by Tukey’s multiple comparison test. All values ae expressed as the mean ± standard error (S.E.); *p* < 0.05 was considered to indicate a statistically significant difference.

## 3. Results

### 3.1. Drug Delivery to the Lens by Instillation of LAN-ONSs and NIL-ONSs

Figure 1A,D show images of the LAN-ONSs and NIL-ONSs, while Figure 1C,F show the size frequencies of drug particles in the LAN-ONSs (Figure 1C) and NIL-ONSs (Figure 1F). Both of the LAN-ONSs and NIL-ONSs appeared to be cloudy. The particle sizes of LAN-ONSs and NIL-ONSs were 108.8 ± 10.8 nm and 89 ± 9 nm, respectively. Moreover, the LAN and NIL particles were nearly spherical in shape (see Figure 1B,E). Figure 2A,B show the drug content in the lenses of rats instilled with LAN-ONSs and NIL-ONSs. Both LAN and NIL were detected in the lens for 24 h after instillation. On the other hand, the changes in LAN and NIL were not observed in the non-instilled eye (left eye).

### 3.2. Improvement of Lens Opacification in Sodium-Selenite-Induced Cataractic Rats by Repeated Co-Instillation of LAN-ONSs and NIL-ONSs

Figure 3 shows the change in lens opacification of rats after injection with sodium selenite. In the nucleus of the lens of both eyes, the onset of lens opacification began 3 days after a single injection with sodium selenite, and opacification was still observed at 35 days after the injection. The opacity level peaked at approximately 7 days after the injection of sodium selenite; after that, the opacity level gradually decreased with time. Figure 4 shows the effects of LAN-ONSs (A and B) and NIL-ONSs (A and C) on lens opacification in the selenite-induced cataractic rats. No improvement in lens opacification was observed in the rats instilled with LAN-ONSs and NIL-ONSs individually, as the opacity levels in the rats instilled with ophthalmic formulations were similar to those in the corresponding selenite-induced cataractic rats instilled with saline (saline). Next, we tested whether the combination of LAN-ONSs and NIL-ONSs attenuated the lens opacification caused by the injection of sodium selenite (Figure 5). The combination of LAN-ONSs and NIL-ONSs reduced the opacity levels in sodium-selenite-induced cataractic rats, with the opacity levels being up to 43% reduced in comparison with the saline group.

### 3.3. Changes in the Cataract-Related Factors of Sodium-Selenite-Induced Cataractic Rat by the Co-Instillation of LAN-ONSs and NIL-ONSs

Figure 6 shows the effect of LAN-ONSs and NIL-ONSs on cataract-related factors in the lenses of sodium-selenite-induced cataractic rats co-instilled with LAN-ONSs and NIL-ONSs. The Ca^2+^ contents and calpain activity in the sodium-selenite-induced cataractic rats were significantly higher in comparison with the normal rats, and the Ca^2+^–ATPase activity was 55.3% that of normal rats. On the other hand, the co-instillation of LAN-ONSs and NIL-ONSs attenuated the enhancement of Ca^2+^ content and calpain activity; however, the decreased Ca^2+^–ATPase activity was not changed in the sodium-selenite-induced cataractic rat co-instilled with or without LAN-ONSs and NIL-ONSs. The Ca^2+^–ATPase activity, Ca^2+^ contents, and calpain activity in the lenses of rats co-instilled with LAN-ONSs and NIL-ONSs were 90%, 71%, and 65%, respectively, those of the sodium-selenite-induced cataractic rats instilled with saline.

### 3.4. Changes in the Lens Structure of Sodium-Selenite-Induced Cataractic Rat by the Co-Instillation of LAN-ONSs and NIL-ONSs

Figure 7 shows the effect of repeated co-instillation of LAN-ONSs and NIL-ONSs on the lens structure in sodium-selenite-induced cataractic rats. In the lenses of normal rats, sclerotic lens nuclei (i.e., perinuclear refractile rings) were not detected, and the arrangement of lens epithelial cells in the equatorial region of the lens and the position of the lens fiber cell nuclei in the bow area were observed to be normal (Figure 7A,D). In the sodium-selenite-induced cataractic rat repeatedly instilled with saline, the lens nucleus was sclerotic (perinuclear refractile ring) and the areas with eosin staining (yellow dotted line) were increased (Figure 7B,E). In addition, the morphological differentiation into lens fiber cells was delayed (blue arrowhead), and the bow area was wider than in normal rats (arrow, Figure 7E). On the other hand, the lens nucleus area with eosin staining (yellow dotted line) in the sodium-selenite-induced cataractic rat repeatedly co-instilled with LAN-ONSs and NIL-ONSs was smaller than that in the saline instillation, and the width of bow area was similar to that in the normal rats (Figure 7C,F). Furthermore, the number of swollen fibers in the sodium-selenite-induced cataractic rats repeatedly instilled with saline was increased, in comparison with that in normal rats (Figure 7G,H), although the enhanced number of swollen fibers was attenuated with the repeated co-instillation of LAN-ONSs and NIL-ONSs (Figure 7I).

## 4. Discussion

Cataracts are the leading cause of vision dysfunction and blindness worldwide [28]. It has been reported that LAN plays a preventive role against lens opacification, reversing crystalline aggregation in both animals and humans [9]. However, the effect of LAN is insufficient to restore lens transparency, and it has also been reported that the lens opacity does not improve sufficiently with LAN treatment [11]. In this study, we designed ophthalmic nanosuspensions based on LAN and NIL (LAN-ONSs and NIL-ONSs) and found that the combination of LAN-ONSs and NIL-ONSs can rescue the progression of selenite-induced cataracts.

Nanosuspensions have been studied extensively as ophthalmic drug delivery systems (DDSs), and nanosuspension formation has been shown to be a useful technique for improving DDSs with increased ocular bioavailability, targeted delivery, and controlled release in order to overcome diffusion and penetration problems in the ophthalmic field [29,30,31,32,33]. First, we attempted to prepare ophthalmic nanosuspensions based on LAN and NIL in this study. Our previous studies have shown that MC can enhance the crushing efficiency when using bead mill treatment, while HPβCD can prevent particle aggregation in the suspension [34]. In addition, BAC has been widely used as a preservative in the ophthalmic field, while the combination of mannitol reduced the corneal toxicity of BAC [35]. From these previous reports, we selected these reagents (HPβCD, MC, BAC, and mannitol) as additives, and the LAN and NIL were bead-milled. The particle size in the milled LAN and NIL suspensions was found to be nano-size (Figure 1). In addition, as in previous studies, aggregation and degradation in the LAN and NIL nanoparticles was not observed for two weeks after preparation [11,17,25]. Next, we evaluated the drug levels in lens after the instillation of LAN-ONSs and NIL-ONSs (Figure 2). Both LAN and NIL were delivered in the lens and were detected in the lens even 24 h after instillation of LAN-ONSs and NIL-ONSs. These results demonstrate that the instillation of LAN-ONSs and NIL-ONSs may be useful for cataract therapy. Therefore, we used these nanosuspensions (LAN-ONSs and NIL-ONSs) to evaluate whether the LAN–NIL combination can act to restore the lens opacity.

It is important to select appropriate animal models for the evaluation of anti-cataract effects. Many animal models of cataracts have been developed for drug screening. Selenite-induced cataract is an animal model of cataracts produced in suckling rat pups [21,22,23]. This model is useful for experimental studies regarding the biochemical mechanisms of cataract, as it presents several similarities to human cataracts; for example, ascorbate and glutathione concentrations are decreased, and there is suppression of mitosis, cytoskeletal loss, calpain-induced proteolysis, and decreased rate of epithelial cell differentiation [14,24]. Therefore, we used sodium-selenite-induced cataractic rats to evaluate the anti-cataract activity of the ophthalmic formulation proposed in this study. A single subcutaneous injection of sodium selenite led to severe nuclear opacity, and the selenite-induced cataracts in the rat model were observed from 4 days after injection of sodium selenite (Figure 3). Moreover, nuclear opacity was observed for at least 35 days after the injection of sodium selenite (Figure 3). Previous studies have also reported that the posterior subcapsular cataract, swollen fibers, and perinuclear refractile ring were observed at 1, 2–3, and 3 days after the injection, respectively. After that, severe bilateral nuclear cataracts are produced within 4–6 days. Thus, the timing of the onset of the cataracts and the persistence of nuclear cataracts in this study were consistent with previous reports [36].

Next, we investigated the therapeutic effect of LAN in the selenite-induced cataract rat model. Although it has been reported that the interaction of LAN with α-crystallin chaperones increases the ability of these chaperones to restore lens clarity by enhancing their ability to physically dissolve the denatured amyloid-like fibril proteins and protein aggregates present in lenses with cataracts [9], the instillation of LAN-ONSs did not restore the opaque lenses to transparency (Figure 4). We have previously reported that the repeated instillation of LAN-ONSs attenuated the slight collapse of the lens structure in SCR; however, serious structural collapses were not reversed with the instillation of LAN-ONSs [11]. Taken together, in order to provide the advantage of LAN, the instillation of LAN-ONSs may be useful under conditions without the cataract-related factors. In the cataract model, the enhanced Ca^2+^ in the lens promotes the aggregation of α-crystallin through the activation of calpain, and it has been reported that NIL, an anti-hypertensive lipophilic dihydropyridine-type calcium channel blocker, attenuates the enhanced intracellular Ca^2+^ contents, resulting in alleviation of the calpain activation and α-crystallin aggregation [17]. Furthermore, Kuroki et al. have also shown that treatment with NIL protected against toxicity through an anti-oxidative effect [37]. From these findings, we investigated whether combination with NIL-ONSs supports the anti-cataractogenic activity of LAN-ONSs.

It is well-known that the Ca^2+^ uptake during nuclear cataract formation is highest in the nucleus of the selenite cataract, while the Ca^2+^ concentration in the cortex remains lower during this period [38]. Moreover, previous studies have reported that free Ca^2+^ levels in the lens nucleus began to rise at least 2 days before onset of the nuclear cataract [38]. By day 4 post-injection (nuclear cataract), the Ca^2+^ levels are significantly increased, compared to that in a normal lens, as if the calcium-binding sites were saturated. In addition, an important consequence of Ca^2+^ elevation in rodent lenses is the activation of the calcium-activated protease, calpain [38]. On the other hand, the Ca^2+^ contents in the lens were normalized at 35 days after injection of sodium selenite (Ca^2+^ contents 3.9 ± 0.7 µmol/mg protein; *n* = 6).

Therefore, we investigated the effect of the LAN-ONSs and NIL-ONSs on cataract-related factors (Ca^2+^-ATPase activity, Ca^2+^ contents, and calpain activity) in rats with mature cataracts at 7 days (i.e., age when eye drops were initiated) after the injection of sodium selenite. The lenses of rats injected with selenite showed a 55.3% decrease in Ca^2+^–ATPase activity, while the Ca^2+^ content and calpain activity in the lenses of rats injected with selenite were higher than those in normal rats (Figure 6). The co-instillation of LAN-ONSs and NIL-ONSs attenuated these changes in Ca^2+^ contents and calpain activity in the lenses, and the individual instillation of NIL-ONSs also prevented increases in Ca^2+^ content and calpain activity in the lenses (Ca^2+^ contents 6.8 ± 0.7 µmol/mg protein, calpain activity 181 ± 14.8%; *n* = 6). In contrast to these results, the Ca^2+^ contents and calpain activity in the lenses of rats instilled with LAN-ONSs were similar to those instilled with saline (Ca^2+^ contents 8.3 ± 0.9 µmol/mg protein, calpain activity 227 ± 15.7%; *n* = 6). Moreover, neither NIL-ONSs nor LAN-ONSs affected the changes in Ca^2+^–ATPase activity. These results indicated that the NIL-ONSs can regulate the enhanced Ca^2+^ contents, resulting in suppression of calpain activation in sodium-selenite-induced cataractic rats.

On the other hand, no improvement in lens opacification was observed in the rats instilled with NIL-ONSs in this study (Figure 4C); however, the co-instillation of LAN-ONSs and NIL-ONSs reduced the opacity levels in sodium-selenite-induced cataractic rats (Figure 5). In addition, the co-instillation of LAN-ONSs and NIL-ONSs recovered the sclerotic lens nuclei (perinuclear refractile ring), the delay of morphological differentiation, and the number of enhanced swollen fibers in the sodium-selenite-induced cataractic rats (Figure 7). These results suggest that NIL-ONSs supports the anti-cataractogenic activity of LAN-ONSs through the inhibition of Ca^2+^ elevation, and the repeated co-instillation of LAN-ONSs and NIL-ONSs can improve the selenite-induced nuclear opacity in rats.

Further studies are required to demonstrate the mechanism of Ca^2+^ regulation through NIL-ONSs instillation in the lens. Moreover, it is important to evaluate the mechanism of ocular DDSs using LAN-ONSs and NIL-ONSs. We have previously reported that drug nanoparticles in ophthalmic formulations are taken up into the corneal epithelium by energy-dependent endocytosis, such as caveolae-mediated endocytosis, clathrin-mediated endocytosis, and micropinocytosis, resulting in an enhancement of transcorneal penetration [17,25]. Therefore, we plan to demonstrate whether energy-dependent endocytosis relates to the transcorneal penetration and delivery into the lens of LAN-ONSs and NIL-ONSs through the use of inhibitors of energy-dependent endocytosis.

## 5. Conclusions

In the field of nuclear cataract therapy, many medicines, such as LAN, are seemingly promising at the molecular level; however, no pharmacological interventions with proven clinical therapeutic efficiency are available thus far. It is possible that the combination of LAN and NIL may be useful in cataract therapy in the future (Figure 8).

## Figures and Tables

**Figure 1 pharmaceutics-14-01520-f001:**
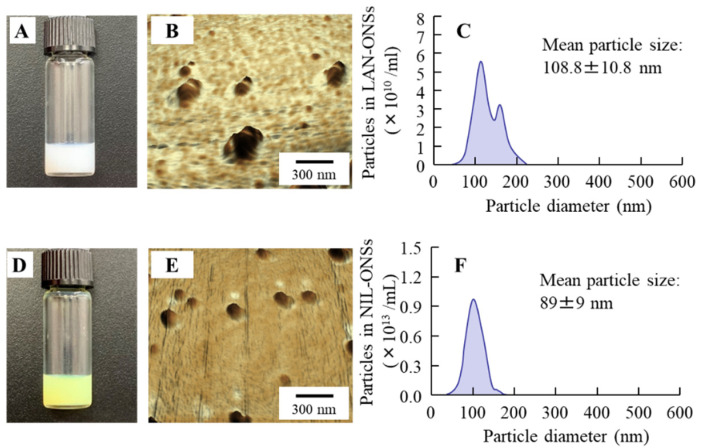
Images and particle sizes of ophthalmic formulations containing LAN and NIL nanoparticles. A and B: digital image (**A**) and AFM image (**B**) of LAN-ONSs. (**C**): particle size frequencies of LAN-ONSs. D and E: digital image (**D**) and AFM image (**E**) of NIL-ONSs. (**F**): particle size frequencies of NIL-ONSs. Bar, 300 nm. Both of the LAN-ONSs and NIL-ONSs appeared cloudy, and the mean particle sizes of the LAN-ONSs and NIL-ONSs were 108.8 nm and 89.0 nm, respectively.

**Figure 2 pharmaceutics-14-01520-f002:**
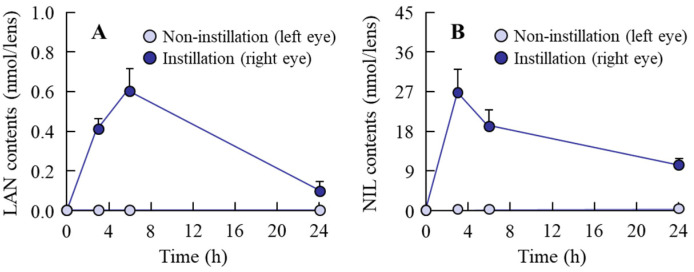
Changes of drug content in the lenses of rats instilled with LAN-ONSs (**A**) and NIL-ONSs (**B**). Twenty microliters of ophthalmic formulations (0.5% LAN or 0.6% NIL), shown in Table 1, were single-instilled into the right eye of 18-day-old rats (*n* = 8). Non-instillation, left eye of rats. Instillation, right eye of rats. The drugs (LAN or NIL) were detected in the lenses after a single instillation of LAN-ONSs or NIL-ONSs.

**Figure 3 pharmaceutics-14-01520-f003:**
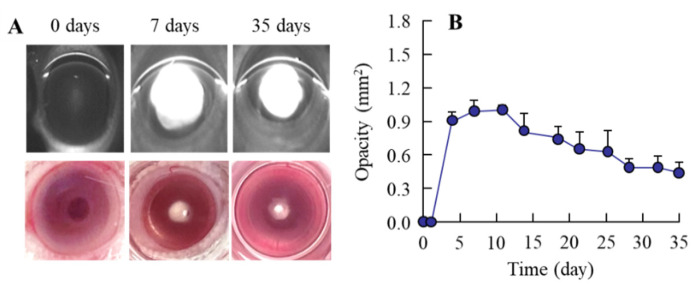
Changes in sodium-selenite-induced opacification in rat lenses. (**A**): digital and Scheimpflug slit images in cataractic rats 0, 7, and 35 days after the injection of sodium selenite. (**B**): changes in opacity levels in the sodium-selenite-induced cataractic rats (*n* = 12). The lens opacification was caused by the injection of sodium selenite, and the opacity level was observed at 3 d after the injection.

**Figure 4 pharmaceutics-14-01520-f004:**
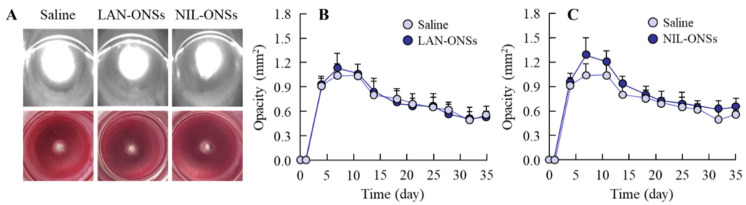
Effect of ophthalmic formulations on opacification in the sodium-selenite-induced cataractic rats. (**A**): digital and Scheimpflug slit images of the sodium-selenite-induced cataractic rats repeatedly instilled with LAN-ONSs and NIL-ONSs. The images were measured at 35 days after the injection of sodium selenite. B and C: changes in opacity levels in the sodium-selenite-induced cataractic rats repeatedly instilled with LAN-ONSs (**B**) and NIL-ONSs (**C**). Twenty microliters of ophthalmic formulations (0.5% LAN or 0.6% NIL), shown in Table 1, were instilled one time/day for 28 days. The start day and the end day of dosing were 7 and 35 days after injection of sodium selenite, respectively. Saline, sodium-selenite-induced cataractic rat repeatedly instilled with saline; LAN- or NIL-ONSs, sodium-selenite-induced cataractic rat repeatedly instilled with LAN-ONSs or NIL-ONSs (*n* = 7–12). The opacity levels were similar between sodium-selenite-induced cataractic rats repeatedly instilled with or without LAN-ONSs and NIL-ONSs.

**Figure 5 pharmaceutics-14-01520-f005:**
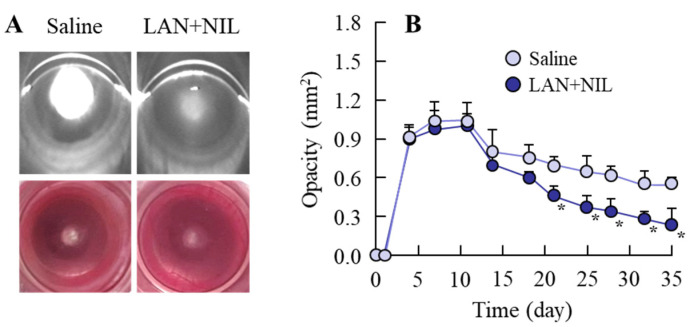
Effect of the combination of LAN-ONSs and NIL-ONSs on opacification in sodium-selenite-induced cataractic rats. (**A**): digital and Scheimpflug slit images of the sodium-selenite-induced cataractic rats repeatedly co-instilled with LAN-ONSs and NIL-ONSs. The images were measured at 35 days after the injection of sodium selenite. (**B**): change in opacity levels in the sodium-selenite-induced cataractic rats repeatedly co-instilled with LAN-ONSs and NIL-ONSs. Twenty microliters of ophthalmic formulations (0.5% LAN or 0.6% NIL), shown in Table 1, were instilled one time/day for 28 days. The start day and the end day of dosing were 7 and 35 days after injection of sodium selenite, respectively. Saline, sodium-selenite-induced cataractic rat repeatedly instilled with saline. LAN + NIL, sodium-selenite-induced cataractic rats repeatedly co-instilled with LAN-ONSs and NIL-ONSs (*n* = 7–12). The combination of LAN-ONSs and NIL-ONSs rescued the opacity levels in the sodium-selenite-induced cataractic rats. * *p* < 0.05 vs. saline for each category.

**Figure 6 pharmaceutics-14-01520-f006:**
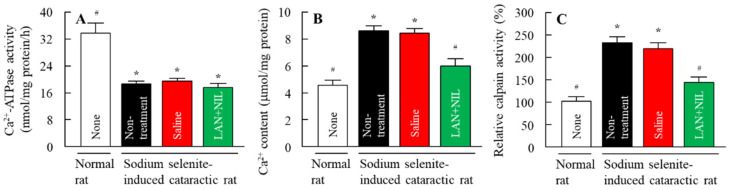
Changes in Ca^2+^–ATPase activity (**A**), Ca^2+^ contents (**B**), and calpain activity (**C**) in the lenses of normal and sodium-selenite-induced cataractic rats co-instilled with LAN-ONSs and NIL-ONSs. The instillation was performed at 7 days after the injection of sodium selenite, and the Ca^2+^–ATPase activity, Ca^2+^ contents, and calpain activity in lenses were measured 3 h after the instillation. None, normal rats. Non-treatment, sodium-selenite-induced cataractic rats. Saline, sodium-selenite-induced cataractic rats instilled with saline. LAN + NIL, sodium-selenite-induced cataractic rats co-instilled with LAN-ONSs and NIL-ONSs (*n* = 6–8). The enhanced Ca^2+^ content and calpain activity in the sodium-selenite-induced cataractic rats was significantly alleviated with the co-instillation of LAN-ONSs and NIL-ONSs. * *p* < 0.05, vs. none (normal rats) for each group. ^#^
*p* < 0.05, vs. saline for each group.

**Figure 7 pharmaceutics-14-01520-f007:**
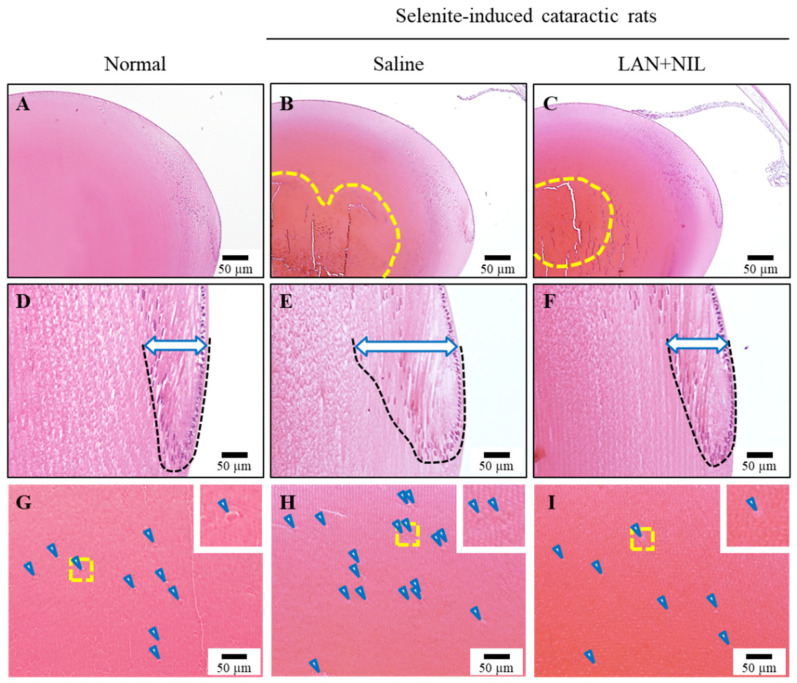
Lens structure of sodium-selenite-induced cataractic rats repeatedly co-instilled with LAN-ONSs and NIL-ONSs. (**A**–**C**): total lens. Yellow area represents the perinuclear refractile ring. (**D**–**F**): equatorial bow region of the lens. Black dotted line and blue arrowhead denote the delay of morphological differentiation. (**G**–**I**): lens fiber area on anterior capsule side. Blue arrowhead denote swollen fibers. Twenty microliters of ophthalmic formulations (0.5% LAN or 0.6% NIL), shown in Table 1, were instilled one time/day for 28 days. The start day and the end day of dosing were 7 and 35 days, respectively. The lens structure was measured at 35 days after the injection of sodium selenite. Scale bar, 50 μm. Normal, lens structure in normal rats. Saline, lens structure in sodium-selenite-induced cataractic rat repeatedly instilled with saline. LAN + NIL, lens structure in sodium-selenite-induced cataractic rat repeatedly co-instilled with LAN-ONSs and NIL-ONSs. The sclerotic lens nucleus (perinuclear refractile ring), the delay of morphological differentiation, and increased number of swollen fibers in sodium-selenite-induced cataractic rats were attenuated by the co-instillation of LAN-ONSs and NIL-ONSs.

**Figure 8 pharmaceutics-14-01520-f008:**
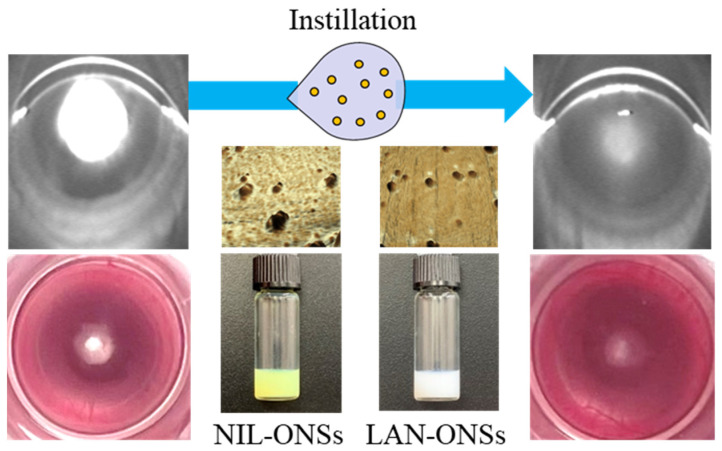
Scheme for recovery of lens opacification in sodium-selenite-induced cataractic rats through the repeated co-instillation of LAN-ONSs and NIL-ONSs.

**Table 1 pharmaceutics-14-01520-t001:** Composition of ophthalmic formulations in rat studies.

Formulation	Composition (%*w*/*v*)						
	LAN	NIL	HPβCD	MC	BAC	Mannitol	Treatment
LAN-ONSs	0.5	–	5	0.5	0.001	0.5	Bead mill
NIL-ONSs	–	0.6	5	0.5	0.001	0.5	Bead mill

## Data Availability

Not applicable.

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
