# Peer review of "Combination of Lanosterol and Nilvadipine Nanosuspensions Rescues Lens Opacification in Selenite-Induced Cataractic Rats"

_pharmaceutics, 2022, doi:10.3390/pharmaceutics14071520_

Round 1

Reviewer 1 Report

Line 21: designed; Line 23: investigated

Introduction:

line 38-39 - revise to "one of the most common cause of blindness worldwide if left untreated: 

line 40-43: to lengthy sentence - break it down in 2

line 45: remove "so on"

line 58-59: redundant to 56-57  

line 70-71: remove "Regularly"

line 73: add a reference (Our previous reports....)

line 79-80: consider removing this sentence

line 95-98:  the formulation instillation information can be moved to a relevant section of the animal studies

line 131: ...the milled dispersions were treated with..." replace "treated" -  may be with subjected to, or similar phrase

Table 1:  title -  add "in" this study;  also consider replacing "this study" with "rat studies" 

Table 1.  "treatment" can be move to the last column;  and can be replaced with "composition (%w/v)"

Section 2.5 Have you assessed drug loading efficiency and stability (physical and chemical) of the nanoparticles?

Section 2.6

- Please add formulation concentration (%w/v), dose (amount of drug/eye), and number of rat as (n=x).   

- Please clarify that two formulations were dosed as separate studies, if that was the case

- texts indicate only one timepoint, 3 hr, which is contrary to the figure 2 that has multiple timepoints

- What was the rationale for reporting LAN levels as a difference between dose and un-dosed eyes?  why this was done only for LAN and not for NIL?  Why not report levels in dosed and un-dosed eye as separate PK curves for both ONSs? 

2.7:  consider revising the heading to "Assessment of Lens Opacification via Imaging".  Please provide information related to dosing, i.e. volume, dose level/eye, frequency (x time/day for y days).  Please describe the method to quantify opacity, i.e. images to graph.   would there a way to assess transparency of the lens which could be a better measure than the area of opacity?

2.8:  line 188-189,  .....following [26]."  please revise

Results: lines 223-226.  avoid redundancy, and keep it concise and clear.

Figure 2:  please report drug levels in dose and un-dosed eyes, if available

3.2 line 240 - revise to "....after a single injection with...."

Figure 3:A. consider adding an image from a control "naive" rat eye.  B. why is opacity reported as mm2?  Does this represent the area of the opaque lens?  

Figure 4 and 5, Legends:  the dosing regimen is not clear. instillation was performed once a day from day 7?  how many total days, and what was the start day and the end day of dosing?  Please clarify these details for all studies. 

3.3 consider a separate section for H.E. /lens structure results

4. Discussion

Overall, this section needs revision.  the flow is not clear.  the paragraphs are lengthy.  Avoid obvious study unrelated statements.  see below comments.

line 351-353:  therefore, delivery...... - revise or remove

page 10:

1st paragraph -  needs significant language/ grammatical revision.  Avoid using drug delivery instead of drug levels in lens.  "...crushed using bead mill treatment..." could be simply "bead-milled"..  It is not clear what authors mean by "repair the lens opacity"?   

2nd paragraph - 1st sentence is not necessary

line 407-442:  a lengthy paragraph - be concise and break it down to 2 or 3 smaller paragraphs.  Avoid redundancy.    

Conclusion:

page 12: this paragraph is a better fit in the Introduction section.  remove from the conclusion, except the last line.  

Author Response

   We carefully revised our manuscript according to the suggestions of the reviewer 1, and details are as follows.

< Q and A for Reviewer 1>

Q1.

Line 21: designed; Line 23: investigated.

line 38-39 -revise to "one of the most common cause of blindness worldwide if left untreated:

line 45: remove "so on"

line 70-71: remove "Regularly"

3.2 line 240 -revise to "....after a single injection with...."

line 351-353: therefore, delivery...... - revise or remove

A1. Thank you very much for pointing this out. In order to respond to the reviewer’s comment, we revised these sentences.

Q2. line 40-43: to lengthy sentence - break it down in 2.

A2. The reviewer’s comment is correct. We broke it down in 2.

Q3. line 58-59: redundant to 56-57.

A3. The reviewer’s comments are very important. We removed the redundant.

Q4. line 73: add a reference (Our previous reports....).

A4. The reviewer’s comment is correct. We add reference (Reference 17).

Q5. line 79-80: consider removing this sentence.

A5. In order to respond to the reviewer’s comment, we removed this sentence.

Q6. line 95-98: the formulation instillation information can be moved to a relevant section of the animal studies.

A6. The reviewer’s comment is correct. We showed the information in the relevant section of the animal studies.

Q7. line 131: ...the milled dispersions were treated with..." replace "treated" -  may be with subjected to, or similar phrase

A7. The reviewer’s comments are very important. We revised this sentence.

Q8. Table 1: title - add "in" this study; also consider replacing "this study" with "rat studies"

A8. In order to respond to the reviewer’s comment, we revised to “in rat studies” (Table 1).

Q9. Table 1. "treatment" can be move to the last column; and can be replaced with "composition (%w/v)"

A9. Thank you for pointing out this. We moved the "treatment" to the last column, and replaced with "composition (%w/v)" (Table 1).

Q10. Section 2.5 Have you assessed drug loading efficiency and stability (physical and chemical) of the nanoparticles?

A10. The reviewer’s comments are very important. Our previous studies showed that the aggregation and degradation in the LAN and NIL nanoparticles did not observed for two weeks after preparation (Ref. 11,17,25). Therefore, we think that the LAN and NIL nanoparticles were stable in the both of physical and chemical. In order to respond to the reviewer’s comment, we added the contents (line 397-398, References 11,17,25).

Q11. Section 2.6 - Please add formulation concentration (%w/v), dose (amount of drug/eye), and number of rat as (n=x).

A11. The reviewer’s comment is correct. The ophthalmic formulation containing 0.5% LAN or 0.6% NIL were used in this study, and the dose (amount of drug/eye), and number of rat were 20 µL, n=8, respectively. In order to respond to the reviewer’s comment, we added the contents (line 165-166).

Q12. Section 2.6 - Please clarify that two formulations were dosed as separate studies, if that was the case

A12. Thank you very much for pointing this out. Two formulations were instilled to same rats, and measured the drug content in the lenses. In order to respond to the reviewer’s comment, we added the contents in the Materials and Methods (line 171-172).

Q13. Section 2.6- texts indicate only one timepoint, 3 hr, which is contrary to the figure 2 that has multiple timepoints

A13. The reviewer’s comment is correct. We revised to “3, 6 and 24 h” (line 168).

Q14. -What was the rationale for reporting LAN levels as a difference between dose and un-dosed eyes? why this was done only for LAN and not for NIL? Why not report levels in dosed and un-dosed eye as separate PK curves for both ONSs?

A14. Thank you very much for pointing this out. Contrast of the nilvadipine, the lanosterol is contained in the normal lens, so the lanosterol levels in the untreated group was measured, and showed as the difference from that in the of non-instilled rats. In order to respond to the reviewer’s comment, we added the contents (line 171-175).

Q15. 2.7: Consider revising the heading to "Assessment of Lens Opacification via Imaging".  Please provide information related to dosing, i.e. volume, dose level/eye, frequency (x time/day for y days). Please describe the method to quantify opacity, i.e. images to graph.   would there a way to assess transparency of the lens which could be a better measure than the area of opacity?

A15. The reviewer’s comments are very important. In order to respond to the reviewer’s comment, we revised the heading to "2.7 Assessment of Lens Opacification via Imaging". The information related to dosing, i.e. volume (0.5% LAN or 0.6% NIL), dose level/eye (20 µL), frequency (1 time/day for 28 days) were mentioned in Material and Method and each Fig legend (the start day and the end day of dosing were 7 and 35 days, respectively, total days 28 days). Transparency of the lens was calculated following: the outline of the lens image was determined by selecting 4 points on the image, and then the transparent area within the outline and thread level were set automatically by the software. The total area of opacity, in pixels, was analyzed by a computer using image analysis software connected to the EAS-1000 system. In order to respond to the reviewer’s comment, we added the contents (line 177-192).

Q16. 2.8: line 188-189, .....following [26]." please revise

A16. The reviewer’s comments are very important. The Ca2+-ATPase activity was calculated following our previous report [26]. We revised this sentence (line 204-205).

Q17. Results: lines 223-226. avoid redundancy, and keep it concise and clear.

A17. The reviewer’s comment is correct. We revised this sentence.

Q18. Figure 2: please report drug levels in dose and un-dosed eyes, if available

A18. Thank you for pointing out this. In order to respond to the reviewer’s comment, we added the data of drug levels in dose and un-dosed eyes (Figure 2).

Q19. Figure 3:A. consider adding an image from a control "naive" rat eye. B. why is opacity reported as mm2? Does this represent the area of the opaque lens?

A19. The reviewer’s comments are very important. In order to respond to the reviewer’s comment, we added the image from a control "naive" rat eye. The mm2 show the area of the opaque lens. We added these contents and image (Figure 3 and Fig. 3 legend).

Q20. Figure 4 and 5, Legends: the dosing regimen is not clear. instillation was performed once a day from day 7? how many total days, and what was the start day and the end day of dosing?  Please clarify these details for all studies.

A20. The reviewer’s comment is correct. The instillation was performed once a day from day 7. The start day and the end day of dosing were 7 and 35 days, respectively (total days 28 days). In order to respond to the reviewer’s comment, we added these contents in the Figure legends (Figure 4 and 5 legend).

Q21. 3.3 consider a separate section for H.E./lens structure results

A21. In order to respond to the reviewer’s comment, we represented the contents (H.E. /lens structure results) in the separate section (3.4) (line 338-339).

Q22. 4. Discussion. Overall, this section needs revision. The flow is not clear. the paragraphs are lengthy. Avoid obvious study unrelated statements. See below comments

A22. Thank you very much for pointing this out. We carefully revised our manuscript according to your suggestions. Thank you for pointing out this.

Q23. page 10:1st paragraph - needs significant language/ grammatical revision. Avoid using drug delivery instead of drug levels in lens. "...crushed using bead mill treatment..." could be simply "bead-milled". It is not clear what authors mean by "repair the lens opacity"?

A23. The reviewer’s comments are very important. In order to respond to the reviewer’s comment, we revised these sentences.

Q24. 2nd paragraph - 1st sentence is not necessary

A24. In order to respond to the reviewer’s comment, we removed the 1st sentence in the 2nd paragraph.

Q25. line 407-442: a lengthy paragraph - be concise and break it down to 2 or 3 smaller paragraphs. Avoid redundancy.

A25. Thank you very much for pointing this out. We revised the sentence, and boke it down to 2 smaller paragraphs.

Q26. Conclusion: page 12: this paragraph is a better fit in the Introduction section.  Remove from the conclusion, except the last line.

A26. The reviewer’s comments are very important. We moved these sentences from Conclusion.

Thank you for great comments.

Reviewer 2 Report

The authors describe in their manuscript entitled "Combination of Lanosterol and Nilvadipine Nanosuspensions  Rescues Lens Opacification in Selenite-Induced Cataractic Rats" a very interesting topic and an unsolved need in medicine.

The manuscript is well written and the research design is very well choosen.

I would recommend some minor changes: to improve the quality of the manuscript.

- the experimental section contains many citations to previus work of the author´s where experimental details can be found. I would suggest, to include the protocols also in this work, this makes it for the reader much easier to follow the method instead of searching the protocol in different citations.

- Introduction: line 45: "..... , and so on..."  the expression "and so on" does not sound very well ; it should be removed or substituted by an other phrase

I recommend for publication.

Author Response

   We carefully revised our manuscript according to the suggestions of the reviewer 2, and details are as follows.

< Q and A for Reviewer 2>

Q1. The experimental section contains many citations to previous work of the author´s where experimental details can be found. I would suggest, to include the protocols also in this work, this makes it for the reader much easier to follow the method instead of searching the protocol in different citations.

A1. The reviewer’s comment is correct. We added the protocols in this work. Thank you very much for pointing this out (line 165-166, 171-172, 174-175, 177-192, 196-197, 205, 219-223).

Q2. Introduction: line 45: "..... , and so on..." the expression "and so on" does not sound very well ; it should be removed or substituted by an other phrase

A2. The reviewer’s comments are very important. In order to respond to the reviewer’s comment, we removed “and so on”.

Thank you for great comments.
